# The Molecular Basis of the Augmented Cardiovascular Risk in Offspring of Mothers with Hypertensive Disorders of Pregnancy

**DOI:** 10.3390/ijms25105455

**Published:** 2024-05-17

**Authors:** Asimenia Svigkou, Vasiliki Katsi, Vasilios G. Kordalis, Konstantinos Tsioufis

**Affiliations:** 1Independent Researcher, 163 42 Athens, Greece; asimeniasvigkou@gmail.com; 2Cardiology Department, School of Medicine, Hippokration General Hospital, National and Kapodistrian University of Athens, 157 72 Athens, Greece; kptsioufis@gmail.com; 3School of Medicine, Aristotle University of Thessaloniki, 541 24 Thessaloniki, Greece; kordalisvg@gmail.com

**Keywords:** preeclampsia, atherosclerosis, endothelial dysfunction, cardiometabolic risk factors

## Abstract

The review examines the impact of maternal preeclampsia (PE) on the cardiometabolic and cardiovascular health of offspring. PE, a hypertensive disorder of pregnancy, is responsible for 2 to 8% of pregnancy-related complications. It significantly contributes to adverse outcomes for their infants, affecting the time of birth, the birth weight, and cardiometabolic risk factors such as blood pressure, body mass index (BMI), abdominal obesity, lipid profiles, glucose, and insulin. Exposure to PE in utero predisposes offspring to an increased risk of cardiometabolic diseases (CMD) and cardiovascular diseases (CVD) through mechanisms that are not fully understood. The incidence of CMD and CVD is constantly increasing, whereas CVD is the main cause of morbidity and mortality globally. A complex interplay of genes, environment, and developmental programming is a plausible explanation for the development of endothelial dysfunction, which leads to atherosclerosis and CVD. The underlying molecular mechanisms are angiogenic imbalance, inflammation, alterations in the renin–angiotensin–aldosterone system (RAAS), endothelium-derived components, serotonin dysregulation, oxidative stress, and activation of both the hypothalamic–pituitary–adrenal axis and hypothalamic–pituitary–gonadal axis. Moreover, the potential role of epigenetic factors, such as DNA methylation and microRNAs as mediators of these effects is emphasized, suggesting avenues for future research and therapeutic interventions.

## 1. Introduction

Hypertensive disorders of pregnancy affect approximately 5% to 10% of pregnant women [1]. Preeclampsia (PE) is responsible for 2 to 8% of pregnancy-related complications, greater than 50,000 maternal deaths, and over 500,000 fetal deaths worldwide [2]. According to the current International Society for the Study of Hypertension in Pregnancy guidelines, PE is new onset of hypertension (systolic blood pressure (SBP) > 140 mmHg or diastolic blood pressure (DBP) > 90 mmHg on two occasions at least 4 h apart in a previously normotensive patient) at or after 20 weeks of pregnancy and the coexistence of 1 or more of the following new-onset conditions: significant proteinuria, maternal organ dysfunction, or uteroplacental dysfunction [3]. Diagnostic criteria of proteinuria are proteinuria of greater than or equal to 0.3 g in a 24 h urine specimen, a protein (mg/dL)/creatinine (mg/dL) ratio of 0.3 or higher, or a urine dipstick protein of 1+ [4]. Preeclamptic women may have no noticeable symptoms and the first signs of PE are often detected during routine prenatal visits. Otherwise, PE is present with headache usually frontal, generalized edema, vision disturbance, right upper quadrant or epigastric pain, amnesia, and other mental status changes [4]. PE is associated with an increased postpartum risk of cardiometabolic (CMD) and cardiovascular disease (CVD) in both women and their offspring, although the underlying mechanisms are still not fully explained [5].

The main pathophysiology of PE is severe vasospasm due to endothelial damage that leads to abnormal blood flow [6]. Shortly after fertilization, maternal and embryonic signals prepare the uterus for successful implantation and placental development [7]. In normal pregnancies, trophoblast cells undergo invasion into the maternal decidua, resulting in obliteration of the tunica media of myometrial spiral arteries and spiral artery remodeling [8]. In PE, on the other hand, the decidualized endometrium is poorly invaded by trophoblasts. Spiral arteries lose vessel surface area and are unable to remodel and expand resulting in low circulating blood volume and high vascular resistance [9]. Maternal cardiovascular risk factors such as obesity, diabetes, smoking, and hypertension can impair the vascular development and function of the placenta [7]. The decrease in uteroplacental blood flow causes placental ischemia and thus fetal growth impairment because the fetus is deprived of important nutrients and oxygen necessary for development [9]. Placental insufficiency, endothelial dysfunction, and hypoxia are responsible for the release of several factors that play a critical role in the pathogenesis of PE [10]. Biomolecules involved in inflammation, oxidative stress, and angiogenesis are related to the pathogenesis of future cardiovascular problems in offspring [5].

The incidence of cardiometabolic and cardiovascular diseases is constantly increasing [11]. Although cardiovascular diseases are thought to affect only adulthood, initial vascular alterations are detectable early in life [12]. Atherosclerosis, the major cause of acquired CVD begins in childhood and is related to several risk factors [13]. CVD is globally the main cause of morbidity and mortality. In 2015, the World Health Organization (WHO) estimated that more than 17.7 million deaths, representing a total 31% of deaths in the whole world were due to CVD [14]. Cardiometabolic risk factors that are associated with CVD are blood pressure, body mass index (BMI), abdominal obesity, total fat mass, cholesterol, triglycerides, and insulin [15]. A study from the National Health and Nutrition Examination Survey from 2001 to 2016, in which 12,000 individuals ages 12 to 19 participated, showed that 1 in 7 had high blood pressure [16]. Another worrying fact is that according to the WHO, 40 million children aged <5 years and more than 330 million children aged 5–19 years were overweight or obese [11]. Many studies agree that PE might have an impact on metabolic and biochemical outcomes in offspring [17]. The goal of this review is to examine how pregnancies complicated by PE might affect the likelihood of CVD in the children born from those pregnancies.

## 2. Effects on Offspring

### 2.1. Prematurity

Prematurity is when a baby is born between 22 and 37 weeks of gestation [18]. Prematurity affects approximately 10% of the general population, with PE accounting for 36% of these cases [19] PE stands as a primary iatrogenic factor contributing to preterm labor, given that delivery remains the definitive treatment for this condition, especially when PE is severe. Otherwise, healthcare providers recommend frequent visits to monitor the blood pressure and the health of the fetus [20]. Prematurity is responsible for neonatal morbidity and mortality worldwide [19]. Premature infants are in danger of chronic diseases, such as CMD and CVD. These adverse outcomes increase as the gestational age at birth decreases [21].

### 2.2. Low Birth Weight

Fetuses born to preeclamptic mothers result in small for gestational age (SGA) neonates [9,19]. PE is associated with placental insufficiency and restricted blood flow to the placenta and fetus, conditions that can increase the likelihood of reduced fetal growth and thus the birth of SGA babies. Premature labor as a consequence of PE is also a risk factor for SGA infants [22]. The risk of SGA newborns is four times higher after in utero exposure to PE [19]. Odegard et al. noticed that offspring exposed to PE had a 5% decrease in birth weight and this reduction was ever greater in cases of severe and early onset PE [22]. Low birth weight is a risk factor for CVD [9].

### 2.3. Cardiometabolic Risk Factors

Searching PubMed, there are two meta-analyses that compare cardiometabolic risk factors in individuals exposed to PE in utero with the control group [23,24]. The meta-analysis of Wang et al. includes 16 case–control studies from 1 January 2010 to 31 December 2019. From all cases, 4046 were in the experimental group and 31,505 were in the control group [23]. The meta-analysis of Andraweera et al. includes 24 prospective and retrospective full-text reviews until 4 June 2018 [24]. The cardiometabolic risk factors that were analyzed by both meta-analyses were SBP, DBP, BMI, total cholesterol, low-density lipoprotein cholesterol (LDL), high-density lipoprotein cholesterol (HDL), non-HDL cholesterol, triglycerides, glucose, and insulin [23,24]. The results of meta-analyses are presented in Table 1 and Table 2. As shown in the Tables, some results from the two meta-analyses coincide, while some do not. Wang et al. noticed that SBP, DBP, BMI, total cholesterol, HDL, and non-HDL were elevated in offspring exposed to PE, whereas triglycerides, glucose and insulin were decreased relative to those who were not exposed to PE. The levels of LDL were almost similar between the two groups [23]. On the other hand, Andraweera et al. found no significant difference in either lipid, glucose, or insulin levels. They showed that offspring of preeclamptic pregnancies had 5.17 mmHg greater SBP, 4.06 mmHg greater DBP, and 0.36 kg/m^2^ greater BMI compared to offspring of control pregnancies [24]. The increased BMI was mainly observed in later life [9]. Wang et al. noticed that the waist circumference of offspring exposed to PE was also significantly increased compared to offspring who were not exposed to PE with a mean difference of 1.37 cm [25]. Abdominal obesity is a risk factor for premature atherosclerosis and CVD [26]. Visceral adiposity correlates with both a hyperlipolytic state resistant to insulin and a disrupted release of adipokines including inflammatory cytokines [27]. Adipocytes accumulated in the abdominal can generate tumor necrosis factor alpha (TNF-a), a potent inflammatory cytokine, establishing an association between adipose tissue and inflammation [28]. Therefore, waist circumference should be adopted as a routine measurement in examination alongside BMI to classify the CVD risk [27].

Bi et al. analyzed randomized clinical trials, cohort, and case–control studies from inception to June 2021 in order to evaluate the long-term impact of PE on adolescents. They found that mean arterial blood pressure, SBP, and DBP were elevated in puberty among offspring exposed to PE. These offspring were also at augmented risk of obesity with a higher BMI from 10 years of age. However, they did not discover any relationship between PE and levels of total cholesterol, LDL, HDL, triglycerides, glucose, and insulin in puberty [17]. All these findings confirm that children born to preeclamptic women have a noticeable increment in blood pressure and BMI later in their lives. Various researchers, among them, Stadler et al., noticed that total cholesterol and non-HDL cholesterol are augmented in cord blood in offspring exposed to early-onset PE, whereas triglycerides and HDL were not significantly different [29]. It seems that factors during pregnancy and labor may influence the results in cord blood [24]. However, there is no clear correlation between PE and lipid levels, glucose, and insulin yet [17,23,24].

### 2.4. Structural Alterations

Neonates born to pregnancies complicated by PE have vascular endothelial dysfunction. Yu et al. noticed that offspring exposed to PE had a reduction twice as large of total vessel density in the skin [30]. Augmentation index (Aix), suprasystolic pulse pressure (ssPP), and microvascular function are novel risk factors for CVD. Both Aix and ssPP are used as a non-invasive measure of vascular stiffness. Aix is connected with CVD risk, while ssPP is associated with obesity [19,31]. Microvascular function is measured using the two parameters of peak perfusion: time to max (TM) and recovery time (time to half, TH2) with laser Doppler perfusion monitoring [19]. Plummer et al. studied the hemodynamic profiles of children born from preeclamptic women aged 8–10 years. They observed that Aix, ssPP, TM, and TH2 were significantly augmented, while there was no difference in peak perfusion. These findings suggest that the impairment of microcirculation results in less compliant large vessels, delay in the endothelial-independent myogenic response, and impaired vasodilation post-ischemia [31].

PE in combination with prematurity and low birth weight is also associated with changes in the number of both nephrons and cardiomyocytes [9,32]. The reduction in the number of nephrons leads to low rates of renal filtration, which in turn is associated with high blood circulating volume and hypertension [33]. Moreover, increased blood pressure may be the result of glomerular hypertrophy and decreased renal vascular dilation [9]. On the other hand, the reduction in the number of cardiomyocytes and their accelerated maturations result in abnormal hypertrophy [34]. Changes in cardiac structure, such as greater wall thickness affect the left end-diastolic volume and it is a sign of premature myocardial disease [35]. Cetinkaya et al. found that premature infants exposed to PE had left ventricle diastolic dysfunction, resulting in CVD risk [36]. Although the majority of studies that have investigated nephron and cardiomyocyte numbers have been performed in animals, human studies appear to support these data as offspring exposed to placental insufficiency have an increased risk of end-stage kidney disease and altered heart chamber anatomical characteristics compared to controls [9].

## 3. Mechanisms Linking Maternal Preeclampsia and Offspring CVD

The relationship between maternal PE and increased risk of CVD in offspring is well known [37]. However, the mechanisms underlying this association have not yet been fully elucidated. Studies agree that a complex interplay of shared genetic factors, family environmental factors, and developmental programming may be the cause [38,39]. Furthermore, the role of epigenetics is also recognized as a contributor to this association [39]. The potential mechanisms that link maternal PE to CVD in offspring are shown in Figure 1.

### 3.1. Shared Genetic Factors

An increasing amount of research is being conducted on the relationship between genetics and PE [40]. The incidence of PE is higher among women who were born of a pregnancy complicated by PE, suggesting that PE is heritable in 31% of cases [8]. Shared alleles that predispose individuals to vascular diseases are more prevalent among women with PE, men who father pregnancies complicated by PE, and offspring born from preeclamptic women [37]. Genetics might contribute to the link between maternal PE and the risk of CVD in offspring [38]. Some offspring who experience CVD in adult life after exposure to PE in utero may be genetically susceptible to CVD and these individuals may experience CVD even if they had not been exposed to PE in utero [37]. Specific genetic variations might predispose individuals to various vascular conditions, with some experiencing PE during pregnancy and others developing CVD [39].

Advances in genetic technology with exome sequencing and clinical genome have a major impact on individualized medicine [41]. An example of this technology was the genetic dissection of chromosome 2 and the identification of four single nucleotide polymorphisms (SNPs) on locus 2q22 [8]. These independent SNPs within four genes associated with both PE and CVD were lactase (LCT, rs2322659), low-density lipoprotein receptor-related protein 1B (LRP1B, rs35821928), rho family GTPase 3 (RND3, rs115015150), and grancalcin (GCA, rs17783344) [37]. Løset et al. also discovered a relationship between SNPs and CVD risk factors, such as glucose levels, triglycerides, and body weight [42]. These SNPs were found in samples from both mothers with preeclampsia and their adolescent offspring [37]. Sitras et al., in a meta-analysis, studied the transcriptome based on data from placental tissue samples from preeclamptic women and gene expression profiles from cardiovascular patient blood samples and found that 22 genes were common for PE and CVD [43]. These genes were related to components that were linked to PE and CVD and included chemokines, inflammation-mediated cytokines, interleukin signaling, oxidative stress, and B-cell activation [8].

### 3.2. Shared Environment

Shared lifestyle between mother and offspring may have an impact on the augmented risk of offspring CVD and maternal PE similar to shared genes [37]. Environmental factors, such as unhealthy diets, sedentary lifestyles, lack of exercise, low socio-economic status, and low educational levels are associated with both maternal PE and CVD [44]. In a study of 369 infants whose mothers smoked during pregnancy, it was found that the number of cigarettes smoked per day was associated with the severity of hypertension in offspring [45]. All these adverse lifestyle conditions are proposed to act as “second hits”. It means that although some individuals are programmed for increased risk, they develop disease only when exposed to a “second hit” later in life [46]. Pregnancy itself can be viewed as a “second hit” in the series of events leading to PE and subsequently to CVD later in life for both the mother and her offspring [44].

Pregnant women are exposed to a variety of endocrine-disrupting chemicals (EDCs), which are substances that can interfere with the endocrine system, potentially leading to adverse effects on human health, including in the cardiovascular system. These chemicals have the ability to interfere with endogenous hormone action through various molecular mechanisms [47]. EDCs can reach the placenta through maternal blood and since the placenta is not completely effective as a protective barrier against EDCs, they bioaccumulate and reach the fetus [48]. In recent years, the variety of EDCs has been steadily growing, resulting in a heterogeneous group with diverse properties and effects [49]. Phytoestrogens interact with estrogen receptors and harm the human placenta by influencing primary human endometrial stromal cells [50]. Polybrominated diphenyl ethers (PBDEs) and perfluoroalkyl substances (PFAS) potentially affect the activity of human chorionic gonadotrophin (hCG), the thyroid hormone production, the differentiation and invasion of cytotrophoblasts (CTBs) and angiogenesis [47]. Phthalates exhibit anti-estrogenic, anti-androgenic, anti-progestogenic, and anti-thyroid properties. They can disrupt intracellular signaling pathways by interacting with nuclear receptors, hormone receptors, transcription factors, and ion channels [51]. Some studies indicate that polychlorinated biphenyls (PCBs) act as aryl hydrocarbon receptor (AHR) agonists, while others suggest they can modulate estrogen and thyroid hormone system activities. They can trigger apoptosis of trophoblast cells, cause proliferative changes in human trophoblast cell lines, and induce antiangiogenic effects at the maternal–fetal interface [52]. All these data suggest that EDCs affect endothelial function through various cellular mechanisms, potentially leading to endothelial dysfunction. These alterations could ultimately contribute to the development of CVD, particularly atherosclerosis, by serving as underlying factors and risk indicators [49].

### 3.3. Developmental Programming

In the 1980s, David Barker studied the link between exposure to negative conditions in utero during critical periods of rapid growth and later life disease. He proposed that an adverse uterine environment has an effect on CMD and CVD in adult life. He derived the Barker hypothesis, which was originally referred to as the “fetal origins of adult disease” and now is called the Developmental Origins of Health and Disease hypothesis [53,54]. While most of the initial work was on intrauterine undernutrition, it was also found that intrauterine exposure to PE and its results such as intrauterine growth restriction [IUGR] and prematurity have an impact on the development of hypertension, coronary heart disease and non-insulin-dependent diabetes in offspring [9]. Fetuses respond by modifying patterns of gene expression, leading to alterations in metabolism and stress response mechanisms [29]. Barker’s hypothesis is a paradigm of the role of developmental programming in the relationship between PE and the risk of CVD in offspring [8].

Several studies propose that the augmented CVD risk in infants may be a long-term consequence of fetal exposure to PE and cannot be explained only by shared genes and lifestyle [31]. Vascular dysfunction also participates in the complex pathogenesis of PE and CVD [55]. Jayet et al. observed that offspring born from preeclamptic women had approximately 30% higher pulmonary artery pressure and impaired vascular function while their siblings born after a normotensive pregnancy displayed normal vascular function [56]. However, if developmental programming was the only cause of the correlation between offspring CVD and maternal PE, then only individuals exposed to PE in utero would display adverse cardiovascular profiles, and siblings born from normotensive pregnancies would not be affected [31]. The HUNT study (Nord-Trøndelag Health Study) consists of three population-based surveys comparing CVD risk factors of adult offspring born after hypertensive pregnancies with those born after normotensive pregnancies. Young adults exposed to hypertensive disorders in pregnancy had higher SBP and DBP, an increase in BMI and wilder waist circumference compared to offspring of normotensive pregnancies. Intriguingly, their siblings born after a normotensive pregnancy had an identical risk profile. This suggests that shared genetic components or shared environment may contribute to this association. However, this study cannot exclude the effects of intrauterine exposure on maternal hypertension [57].

### 3.4. Epigenetics

Epigenetics refers to modifications in gene expression, without altering the underlying DNA sequence. Epigenetic changes include DNA methylation, non-coding RNA (ncRNA), and histone modifications, which often result in chromatin remodeling. They have been investigated within developmental processes due to their ability to regulate abnormal gene expression [58]. Environmental exposures can induce epigenetic modifications, offering potential insights into whether early-life adverse exposures are linked to future risks of chronic diseases [31]. Epigenetic changes have an impact on placental development by various mechanisms [59]. Wingless-related integration site family, member 2 (*Wnt2*) belongs to the *Wnt* gene family, which comprises genes sharing structural similarities and encoding secreted signaling proteins crucial in the *Wnt* signaling pathway. These proteins play significant roles in various developmental processes, such as regulating cell fate and patterning during embryogenesis. Studies in rats have shown that increased methylation of the *Wnt2* results in decreased expression levels of the *Wnt2* gene, suggesting a potential role in impaired trophoblast invasion and poor remodeling of the spiral arteries [60]. Placental dysfunction may induce epigenetic alterations in umbilical cord blood, leading to an augmented risk of CVD in offspring [61]. There is strong evidence that epigenetic processes are associated with PE and CVD and may be inherited in offspring [62]. Thus, epigenetic modifications are commonly used as diagnostic and prognostic biomarkers for adverse outcomes [58]. Developmental programming can also be affected, changed, and reprogrammed. Modifications of enzymes and DNA, transcription factors, and oxidative stress markers can alter a newborn’s sensitivity to angiotensin (Ang) II, augmentation of the renin–angiotensin–aldosterone system (RAAS), nephron number, and organ development, which can all lead to hypertension [63].

## 4. Molecular Mechanisms

Inflammatory cytokines, reactive oxygen species (ROS), and antiangiogenic factors are released in PE due to placental ischemia. All these factors target endothelial cells and are responsible for the release of vasoactive substances and the decrease in vasodilators, leading to vasoconstriction and hypertension [64]. Intrauterine exposure to this environment is related to the increased risk of CVD and hypertension later in life in offspring. Angiogenic imbalance, inflammation, alterations in the RAAS, sympathetic nervous system (SNS), endothelial and other mediators, oxidative stress, increased glucocorticoids, and sex hormones are some of the mechanisms that explain this link [10,65]. Epigenetic changes consisting of DNA methylation and ncRNA are also important moderators of the relationship between maternal PE and CVD risk [59].

### 4.1. Angiogenic Imbalance

Angiogenic disparity is implicated in PE. Syncytiotrophoblast stress due to placental ischemia and hypoxia releases antiangiogenic components, like soluble fms-like tyrosine kinase-1 (sFlt-1) and soluble endoglin (sEng) [66]. Both sFlt-1 and sEng advocate vascular dysfunction and capillary permeability [24]. They antagonize the proangiogenic molecules placental growth factor (PlGF) and vascular endothelial growth factor (VEGF) by binding to them and inhibiting their interaction with their cell surface receptors [67]. The decline of PIGF and VEGF is associated with a vasoconstrictive state, microemboli and endothelial dysfunction [67]. Byers et al. studied rodents and noticed that overexpression of sFlt-1 was associated with impaired vascular function in offspring [68]. Another study in rats by Lu et al. demonstrated that sFlt-1 resulted in augmented SBP and DBP, but only in male offspring [69]. Angiogenic imbalance is known to exist in adulthood in offspring exposed to PE and is associated with increased blood pressure [70]. It is also involved in the development of diastolic dysfunction and in heart defects, leading to heart failure [71,72]. Elevated plasma levels of sFlt-1 suggest sex-specific differences in the developmental programming of glucose metabolism, with female offspring having altered glucose metabolism [73]. McDonnold et al. performed intraperitoneal glucose tolerance testing in mice that were injected through the tail vein with adenovirus carrying sFlt-1 and noticed that female offspring had higher fasting and peak glucose values [74]. The mechanisms of sex-specific offspring outcomes are not clear, but sexually placental dimorphic adaptations can be a possible reason [73]. Regrettably, the impacts of elevated levels of sEng on offspring health remain unclear. Experiments conducted on mice in vivo have demonstrated that sEng leads to increased vascular resistance, and consequently to an increase in blood pressure, and may be involved in glucose metabolism [75].

### 4.2. Inflammation

Inflammation plays a significant role in the pathogenesis of PE and CVD in newborns [8]. Placental ischemia and hypoxia lead to an augment in inflammatory cytokines and cells, such as T helper (Th) 1 and Th 17 cells, TNF-a, Natural Killer (NK) cells and Angiotensin II Type 1 Receptor Agonistic Autoantibody (AT1-AAs) [76]. In normal pregnancies, the Th2 phenotype predominates in order to suppress excessive inflammation. On the contrary, in PE, Th cells shift towards the Th1 phenotype. The release of proinflammatory cytokines, such as interleukin (IL)-18 and IL-12 is augmented, leading to apoptosis and impaired trophoblast invasion [77]. Intrauterine exposure to inflammation is related to various changes in the fetal immune system. Guillemette et al. found that newborns from preeclamptic mothers had higher levels of TNF-a at birth [78]. Furthermore, Hu et al. demonstrated that there is a decline in regulatory T cells (Tregs), leading to a reduced capacity to dampen inflammation that persists into early childhood [79]. Inflammation has a great impact on the pathogenesis of atherosclerosis, leading to CVD [79,80]. Zhang et al. studied the influence of placental hypoxia on large blood vessels in offspring rats [80]. Hypoxia activates hypoxia-inducible factor 1 (HIF-1). HIB-1a stimulates NFκB through the increased IKKβ kinase expression and liberation of NFκB to the nucleus [81]. Activated NFκB releases pro-inflammatory cytokines IL-1β and TNF-a, which in turn leads to the development of atherosclerosis via endothelium injury and disarrangement of elastic membranes [80,82].

### 4.3. Alterations in the RAAS

RAAS dysregulation is responsible for both the pathogenesis of PE and the long-term programming of hypertension in offspring exposed to PE [83,84]. The most notable components that regulate these processes include the angiotensin-converting enzyme (ACE), Ang II, Ang II type 1 receptor (AT1R) and ACE2/angiotensin-(1–7) (Ang-(1–7))/Mas receptor pathways [84]. In normotensive pregnancies, plasma renin activity (PRA) and aldosterone remain increased. Angiotensinogen, which is cleaved by renin, stimulates Ang I, which in turn increases levels of Ang II through ACE. Ang II then binds to AT1R in order to maintain sodium balance and plasma volume [85]. Ang II is also associated with trophoblast invasion and spiral artery remodeling [86]. Excessive Ang II production may lead to excessive vasoconstriction and negative consequences [87]. Thus, the ACE2/Ang-(1–7) pathway is stimulated to balance increased ACE/Ang II pathway activity. In more detail, ACE2 deconstructs Ang II and Ang-(1–7), acting on its Mas receptor, antagonizes Ang II signaling via AT1R alteration. The maternal systemic vascular resistance is decreased through the ACE2/Ang-(1–7) pathway [83]. The exact role of RAAS in PE remains unclear [88]. The ACE protein expression is augmented in fetal endothelial cells, which enhances Ang II production [89]. It seems that RAAS components, such as increased Ang II and decreased Ang-(1–7) are released into the maternal circulation, resulting in reduced uteroplacental blood flow and abnormal placental development [88]. Furthermore, elevated levels of circulating AT1-AAs have been found in 70–95% of preeclamptic women, compared to 30% of normotensive women [90]. They are directed to a specific epitope on the AT1R and bind to human trophoblasts. There is strong evidence that AT1-AAs have an impact on the pathogenesis of PE through vasoconstriction and aldosterone secretion [91].

Alterations in the RAAS can potentially affect fetal cardiovascular health in the short and long term [92]. Studies in rodents have shown that only male offspring exposed to PE had hypertension during young adulthood [93]. It was observed an augmented Ang II sensitivity in male rats, without any changes in vessel morphology. Castration eliminated the elevated blood pressure response to Ang II, indicating that testosterone may play a part in regulating the sensitivity to Ang II [94]. On the other hand, female offspring developed hypertension after an ovariectomy [93]. Estrogen regulates RAAS activation that occurs during pregnancy by enhancing the expression and activity of ACE2, suggesting their protective role in preventing hypertension [95]. South et al. noticed that teenagers born prematurely tend to have increased RAAS activity, particularly favoring the ACE/Ang II pathway over the ACE2/Ang-(1–7) pathway, which may contribute to higher blood pressure levels. However, they found that this relationship was more pronounced in female and obese individuals [96]. Washburn et al. demonstrated that male adolescents born with a low birth weight from preeclamptic women had increased aldosterone levels and blood pressure compared with those who were born from normotensive women [97].

Moreover, maternal–fetal transfer of AT1-AAs is associated with growth restriction and changes in the structure of organs such as loss of kidney glomeruli, apoptosis in the heart muscle, and infiltration of immature cells in the liver in fetuses. AT1-AAs activate AT1R and contribute to abnormal organogenesis via systemic vasoconstriction and hypoxia [98]. Zhang et al. found that rats immunized with AT1-AAs showed augmented fasting insulin levels, proposing the development of insulin resistance [99]. One possible mechanism is the involvement of AT1R in insulin signaling of beta cells [100]. Nonetheless, further research is necessary to clarify the mechanisms of AT1-AAs-induced fetal metabolic programming [99].

Beyond cardiovascular health, RAAS as a regulator of blood pressure and fluid-electrolyte balance is also related to fetal kidney development [101]. Various perinatal insults, including placental insufficiency, can program kidneys [30]. Since nephrogenesis primarily happens during the third trimester of pregnancy, when PE is also more severe, it is logical to assume that alterations in blood flow and circulatory factors during this critical period may negatively impact fetal renal development [101]. In the initial stages of kidney development, RAAS is upregulated, resulting in the vasoconstriction of renal arteries [102]. Simultaneously, the activity of SNS is also increased, which leads to an augmentation in sympathetic tone in the renal vessels. Vasoconstrictive peptides, such as thromboxane A2 and endothelin are in balance with vasodilator substances such as prostaglandins, nitric oxide (NO), and the Kallikrein–Kinin system [103]. Placental hypoxia in PE results in an unbalanced regulation of vasoactive components. More specifically, there is an increase in vasoconstrictors, a decrease in vasodilators, and an increased vascular sensitivity to Ang II [104]. Both RAAS and SNS have a great impact on hypertension in offspring [101]. Nephron deficiency as a result of PE is associated with impaired blood pressure regulation due to the inability of the kidney to maintain sodium homeostasis and the imbalance in excretory load. Singh et al. showed that reduced excretory capacity could stem from impaired expression of renal sodium transporters and channels [105].

### 4.4. Imbalance of Endothelium-Derived Components

An imbalance of vasodilatory and vasoconstrictive substances causes endothelial dysfunction, which can lead to atherosclerosis, an early stage of CVD [106]. NO is synthesized from L-arginine by nitric oxide synthase (NOS) and triggers vasodilation. NO production can be stimulated by VEGF, which participates in angiogenesis and the proliferation of endothelial cells. A decrease in NO bioactivity is related to endothelial dysfunction [107]. Furthermore, NO production can be augmented by activation of Ang II type 2 receptor (AT2R) and Mas protein [101]. In contrast to the actions of Ang II when it binds to AT1R, it can also bind to AT2R and cause vasodilation through increasing NO [9]. In experimental studies of offspring, administration of either an AT1R or renin receptor blockade inhibitor can prevent the development of hypertension [108]. These treatments decrease asymmetric dimethylarginine (ADMA) and increase AT2R and Mas protein levels in the kidneys [109]. ADMA is an endogenous inhibitor of NO synthase and contributes to the increased risk of hypertension and CVD later in life [110]. Endothelin-1 (ET1) is also associated with endothelial dysfunction through its stimulation of nicotinamide adenine dinucleotide phosphate (NADPH) oxidase-derived reactive oxygen species (ROS) production, which inhibits NO-mediated endothelial relaxation. Additionally, it mediates endothelin A receptors, resulting in the blunting of NO relaxant responses [111]. In PE, both NO and VEGF concentrations are decreased, whereas ET1 levels are increased. During pregnancy, vasoactive components can cross the placenta and affect fetal circulation directly. Disproportionation of endothelium-derived factors in the maternal circulation is also observed in the fetus and may lead to future effects in the offspring [106].

### 4.5. Serotonin Dysregulation

Another vasoactive component that is associated with PE and the risk of CVD in offspring is serotonin (5-HT). In PE, 5-HT is significantly elevated in maternal circulation, placenta, and cord blood, while its degradation is decreased. Monoamine oxidase A (MAO-A) catabolizes 5-HT to its inactive form, 5-HIAA. In the placenta from pregnancies affected by PE, MAO-A exhibits both reduced expression and activity [112]. A decrease in the expression of indoleamine 2, 3-dioxygenase (IDO), a catabolic enzyme, in both the mother and the placenta, which is modulated by inflammation is also observed. Decreased IDO expression leads to the conversion of tryptophan to 5-HT [113]. These can lead to elevated levels of 5-HT within the placenta, and potentially impact the developing fetus [112]. Hyperserotonemia can interact with placental vascular function both directly, through vasoactive mechanisms, and indirectly, via inflammatory pathways, contributing to placental dysfunction in PE [114].

5-HT is readily bound up by intravascular platelets [114]. Endothelial cell injury in PE enhances platelet aggregation, leading to the release of 5-HT and subsequent vasoconstriction. In smooth muscle and the human uterine artery, this vasoconstriction is predominantly mediated by 5-HT receptor 2 (HTR2) [114]. The increased release of 5-HT from platelets alters local vascular function directly by triggering the contraction of vascular smooth muscle and activation of endothelial cells, and indirectly, by amplifying the effects of other vasoactive substances, such as thromboxane A2 and prostaglandins [115].

5-HT also plays a significant role in the pro-inflammatory processes associated with PE. Disturbances in serotonin-immune interactions can lead to abnormalities in the metabolism of the kynurenine pathway, T cell function, and cytokine production. 5-HT binds to T cells and triggers downstream cascades through 5-HTR, which are G protein-coupled receptors. Regulators of G protein signaling control the flow of signals inside the cell and therefore they can affect the production of inflammatory proteins, including cytokines [114]. These proteins have impacts on vascular and platelet dysfunction in the context of PE. Dysfunction in blood vessels and platelets can increase levels of 5-HT in the bloodstream, thereby signaling more to the T cells, amplifying the problem [114]. Various immune cells, including dendritic cells, mast cells, NK cells, T cells, B cells, neutrophils, macrophages, and monocytes, express 5-HTR and enzymes involved in 5-HT signaling. These mechanisms have significant implications for PE, resulting in adverse effects on fetal development [116].

### 4.6. Oxidative Stress

Oxidative stress is an underlying mechanism between fetal insults and the programming of CVD in later life [117]. It is generated when the production and consumption of ROS are imbalanced. Controlled oxidative stress regulates cellular signaling, differentiation, and proliferation [118]. However, it is observed that during PE either oxidative stress markers are elevated, or the antioxidant activity is decreased, leading to the damage of macromolecules [119]. Inflammatory cytokines and AT1-AAs increase ROS production by up to 40% during PE [120]. Placental ischemia itself can also release ROS, such as superoxide and hydrogen peroxide [121]. Increased oxidative stress causes placental oxidative DNA damage, resulting in fetal growth restriction and lipid peroxidation in offspring [122]. Increased levels of protein and lipid oxidation affect cardiometabolic alterations, such as high blood pressure, altered lipid profiles, and insulin resistance [117]. HDL and LDL lipoproteins can undergo oxidative damage and generate highly toxic products, which are transferred to the fetus [123]. Infants born preterm and SGA are particularly vulnerable due to insufficient antioxidants, like vitamins E and A [124]. Hilali et al. found oxidative stress and DNA damage in the cord blood of offspring as they were transferred to the fetus [119].

There is strong evidence that oxidative stress is associated with CMD as beta cells and vascular endothelium are sensitive tissues to oxidative damage [24]. Furthermore, oxidative damage in the kidney, blood vessels, and the heart is commonly acknowledged as a key factor in causing organ dysfunction and atherosclerosis, ultimately resulting in CVD [117]. Studies in animals have shown an increase in renal markers of oxidative stress, many of which were associated with RAAS dysfunction. Bi et al. studied adult sheep exposed to oxidative stress and found that there was an increase in renal 8-isoprostane in response to Ang II and augmented ROS-mediated Ang II responses [125]. Oxidative stress also causes vascular changes, especially endothelial dysfunction [126]. Activation of NADPH pathways results in superoxide production, which can contribute to vascular hyper-reactivity [127]. Thompson and Al-Hasan noticed that superoxide anion reduced vascular NO bioavailability in animals exposed to fetal stress [128]. Additionally, the heart as the organ with the highest oxygen uptake is susceptible to oxidative damage. Stress factors induce cardiac hypertrophy, which is associated with augmented heart NADPH oxidase expression [129].

### 4.7. Activation of the Hypothalamic–Pituitary–Adrenal Axis

Changes in the hypothalamic–pituitary–adrenal (HPA) axis and increased glucocorticoids play a key role in the onset of chronic disease in offspring exposed to PE. Henley et al. observed that a 17-year-old offspring of a preeclamptic mother had increased levels of adrenocorticotropic hormone (ACTH) and cortisol [130]. This suggests the reprogramming of the HPA axis initiated by intrauterine exposure to PE, which persists into adulthood and potentially contributes to elevated blood pressure in offspring affected by PE [130]. Augmentation of the HPA axis is also inversely associated with birth weight. Martinez-Aguayo et al. studied children and adolescents between ages 4 and 16 who were born at low birth weight. They noticed that their aldosterone, cortisol, and blood pressure were increased [131]. Moreover, the relationship between PE and CVD in neonates may be attributed to maternal glucocorticoid metabolism and exogenous glucocorticoid exposure that is often administered in the antenatal period to prevent preterm delivery [132].

### 4.8. Activation of the Hypothalamic–Pituitary–Gonadal Axis

Studies in animals have shown that augmented levels of androgen during pregnancy can lead to hyperactivity of the hypothalamic–pituitary–gonadal axis and modifications in the expression of steroid genes in the gonads of the offspring. The production of testosterone is therefore increased [133]. It is already mentioned that testosterone promotes enhanced Ang II sensitivity [94], which is associated with volume-dependent hypertension [134]. There is strong evidence that prenatal exposure to elevated testosterone levels in preeclamptic mothers is related to fetal growth restriction followed by catch-up growth and increased blood pressure in the female offspring during adulthood [135]. Reduced fetal growth has an established correlation with the risk of CVD and according to Kelishadi et al., catch-up growth is a more significant risk factor for CVD [136]. More et al. observed that prenatal exposure to elevated testosterone was associated with a decrease in the expression of CYP11B2, resulting in a reduction of plasma aldosterone levels. Nevertheless, the plasma volume and the balance between sodium and potassium ions were normal [137]. In female offspring with elevated testosterone levels, plasma concentrations of vasopressin and Ang II, as well as vascular response to Ang II and blood pressure, were all heightened. This could potentially represent a compensatory mechanism aimed at preserving plasma volume, balancing sodium and potassium levels, and therefore regulating blood pressure [135]. Alsnes et al. noticed that all male offspring exposed to preeclampsia and female that were exposed to clinically severe preeclampsia had increased levels of testosterone in early puberty, whereas male adolescents had declines in dehydroepiandrosterone sulfate, testicular volume, and circulating aldosterone. These variations might influence the initiation and advancement of adolescence and potentially contribute to the onset of hypertension and elevated cardiovascular risk. Paradoxically, the female offspring of mothers with mild or moderate preeclampsia had decreased levels of testosterone [138]. Elevated testosterone and smaller testicular volume were also present in adult males who were born after a hypertensive pregnancy [139].

### 4.9. Epigenetic Modifications

Epidemiological studies have demonstrated that epigenetics, when assessed in the placenta or cord blood, may be potential mediators or biomarkers of in utero exposure to PE, as epigenetic modifications occur both in the placenta and in fetal cells. Alterations in important genes during pregnancy may affect the function of the placenta, resulting in an adverse environment for the developing fetus. The fetus adapts to an unfavorable intrauterine environment, resulting in epigenetic changes that mediate CVD risk later in life [61,62].

DNA methylation

The most widely studied epigenetic alteration is DNA methylation. DNA methylation is a biochemical process of adding a cytosine in cytosine-phospho-guanine (CpG) dinucleotide sites, performed by DNA methyltransferases (DNMTs) [31,140]. CpG islands are genomic regions characterized by a high frequency of cytosine and guanine nucleotides being adjacent to each other along the DNA strand and are often targeted by transcription factors [140]. DNA methylation is linked to numerous processes, such as genomic imprinting, X chromosome inactivation, and repression of transposable elements [58,140]. DNA methylation with histone modifications affects the packing of chromatin, influencing the accessibility of transcription factors to the regulatory DNA sequences that control gene expression. Histone modifications include changes to the histone tails, like acetylation, methylation, and phosphorylation [58]. Hypermethylation of CpGs islands and the regulatory regions of promoters are associated with gene repression whereas hypomethylation results in gene activation [141].

Several studies have shown that preeclamptic women have aberrant DNA methylation in the placentas compared to normotensive women, suggesting the role of epigenetics in placental gene modification [14,142]. Wang et al. found that methylation and thus inhibition of CpG island methylator phenotype (CMIP) was associated with elevated expression of VEGFA, VEGFB, and HIF1a [143]. Expression of these genes leads to hypoxia, angiogenic imbalance, and placental insufficiency [144]. Hogg et al. studied the relationship between early-onset PE and the altered methylation of cortisol-signaling genes and steroidogenic genes in the placenta. They noticed that cortisol was elevated only in preeclamptic women [145]. Moreover, Blair et al. examined 20 chorionic villi samples from early-onset preeclamptic placentas and 20 gestational age-matched controls and found 38,840 CpGs sites with important modifications in DNA methylation [146]. Additionally, Brodowski et al. studied the genomic methylation pattern of fetal endothelial colony-forming cells (ECFC) from preeclamptic and normal placentas. They noticed a general loss of CpG methylation in preeclamptic placentas as the majority of CpG sites were hypomethylated in fetal ECFC. Differential methylation patterns of fetal ECFC were found in regions that regulate cell metabolism, transcription, and cell cycle [147]. These data suggest that different methylation patterns in placentas could serve as potential biomarkers for exposure to PE as they are also detected in fetal cells [142,148]. Epigenetics, therefore, provides a potential explanation for the correlation between PE and CVD in offspring and its transmission to future generations [149,150].

An imbalance of endothelium-derived vasodilatory and vasoconstrictive factors is one cause of endothelial dysfunction, which is the early stage of atherosclerosis and CVD. Yu et al. studied the epigenetic regulation of delta-like homolog 1-maternally expressed gene 3 (DLK1-MEG3) region in human umbilical vein endothelial cells (HUVECs), and its connection with endothelium-derived components. They found that the DLK1–MEG3 region was hypermethylated, leading to a decline in NO and VEGF expression, whereas ET1 levels were elevated. This study suggests that methylation of this region may induce endothelial dysfunction and CVD [106]. Genome-wide methylation analysis used neonatal cord blood DNA and showed a notable genome-scale hypomethylation in neonatal cord blood DNA associated with early onset PE, with 51,486 hypomethylated and 12,563 hypermethylated CpGs. DNA modifications were discovered in genes involved in lipid metabolism and inflammation, including IL12B, fatty acid synthase (FAS), phosphatidylinositol 3-kinase 1 (PI3K1), and insulin-like growth factor 1 (IGF1) [151]. Deregulation of both pathways contributes to the increased risk of CVD in offspring [142,151].

Epigenetic modifications to several RAAS gene promoters in neonates can also affect the development of the cardiovascular and renal systems. DNA methylation and histone modifications of the promoter for the AT1R gene, AGTR1, can influence gene expression and potentially affect vascular function as AGTR1 is involved in the regulation of vascular tone and blood pressure [101]. Studies in rats have noticed that PE can result in hypomethylation of AGTR1 and AT1bR overexpression, leading to increased blood pressure and Ang II sensitivity [152]. An important factor limiting the translation of these findings to humans is the genetic disparity between rodents and humans. Rodents have two subtypes of genes transcribing AT1R, while humans have only one, that resembles the AT1aR. Thus, additional investigations are necessary to confirm whether human AGTR1 plays a significant role as a major RAAS gene that is epigenetically targeted by antenatal events [101].

microRNA

Beyond DNA methylation, ncRNAs are another epigenetic modulator. They consist of short ncRNAs like microRNAs (miRNAs) that can suppress transcription and translation or modify protein trafficking and folding [153]. MiRNAs are small single-stranded noncoding posttranscriptional regulatory molecules [154]. They consist of 19–24 nucleotides [155]. They regulate gene expression through base pairing with complementary sequences in their target mRNA [156]. More specifically, 5′ terminal “seed sequence” can combine with the 3′-untranslated region (3′-UTR) of mRNA to degrade mRNA and suppress translation. When miRNA forms a complete base pairing with its target mRNA, it can directly cleave a single phosphodiester bond in the mRNA, leading to mRNA degradation [157]. Their role includes modulating basic cellular activities, such as proliferation, metabolism, immune activities, and apoptosis [158]. MiRNA expression profiles are significantly different between normal and pathological tissues, which could be useful in clinical diagnostics and therapy [159].

Clinical and experimental studies have shown that miRNAs contribute to placental function, affecting angiogenesis, NO production, and trophoblast invasion [142]. They can also mediate RAAS dysregulation [160]. A list of the role of different miRNAs during PE is presented in Table 3 [142,161]. Apart from the development of the placenta, dysregulation of miRNAs affects fetal growth and development, as different miRNA expression patterns can be transferred in offspring [162]. Pan et al. compared 157 miRNAs expression levels in serum exosomes between preeclamptic and normotensive women. They found that 96 miRNAs were upregulated whereas 61 miRNAs were downregulated [158]. These differentially expressed miRNAs are associated with cardiovascular function in offspring. An example of this link is miR-483-5p, which is related to obesity and CVD [163]. Yu et al. demonstrated that the upregulation of miR-146a expression in HUVECs was associated with lower vasculogenic capacity, leading to a decline in microvascular development in the early postnatal period [164]. Moreover, Zhou et al. found that the downregulation of miR-29a/c-3p resulted in impaired fetal endothelial cell immigration through the disturbance of the fibroblast growth factor 2 (FGF2)-stimulated PI3K-AKT1 pathway [165]. These findings confirm that different expressions of miRNAs in fetal endothelial cells are associated with an increased risk of CVD in later life, as impaired angiogenesis is an essential trait of CVD [164,165]. Although there is a link between miRNAs and microvascular dysfunction in the offspring of preeclamptic mothers, the mechanisms have not yet been elucidated [158].

## 5. Conclusions

PE is associated with adverse effects on the cardiometabolic health of offspring, principally blood pressure. On the contrary, there are inconsistencies in studies on the impact of PE on lipid profiles, glucose, and insulin, that require in-depth research [166]. The increased blood pressure in childhood can induce large artery stiffness and worsen arterial function in later life [167]. Children exposed to cardiovascular risk factors are susceptible to developing preclinical atherosclerosis and CVD [168]. The incidence of cardiovascular morbidity is significantly higher among infants born from preeclamptic women [169]. PE is associated with impaired trophoblast invasion and spiral artery remodeling, resulting in restriction in uteroplacental circulation [170]. The intrauterine environment is then marked by multiple insults, including placental insufficiency, ischemia, hypoxia, elevated antiangiogenic factors, oxidative stress, and inflammation [10]. These insults during pregnancy have diverse effects on the fetus, resulting in endothelial dysfunction, abnormal vascular structure, and accelerated atherosclerosis progression [171]. Mechanisms linking exposure to PE in utero and the risk for CVD may be an interaction between shared genes, shared environment, and developmental programming. It is difficult to determine the extent of involvement in the programming of the cardiovascular system as numerous pathophysiological pathways are implicated in the pathogenesis and clinical progression of PE [31,171]. Imbalance of angiogenic factors and endothelium-derived components, dysregulation of 5-HT, inflammation, alterations in the RAAS, oxidative stress, and activation of both HPA and HGA are involved in fetal programming of hypertension and CVD [10,65]. Furthermore, differential DNA methylation patterns and alterations in the expression of miRNAs can affect the above pathways and in turn, contribute to the relationship between PE and CVD [142,143,144].

## 6. Future Directions

The significance of the prenatal environment on the future health of offspring is already known. Since cardiovascular risk factors during pregnancy can directly affect the long-term cardiovascular health of the offspring, enhancing maternal health and the maternal–fetal environment becomes crucial in order to minimize the non-genetic or environmental transmission of cardiovascular risk from one generation to another [92]. The aim is to improve maternal–fetal health and for this reason, studies are needed to examine the effects of the treatment used in preeclamptic women [14]. In additionally, further research is needed to elucidate the effects of PE on the epigenetic changes in the fetus, and whether these alterations persist into adulthood. Understanding the underlying epigenetic mechanisms is challenging in humans due to the requirement for longitudinal cohorts and the need to identify the appropriate tissue for assessment [142].

## Figures and Tables

**Figure 1 ijms-25-05455-f001:**
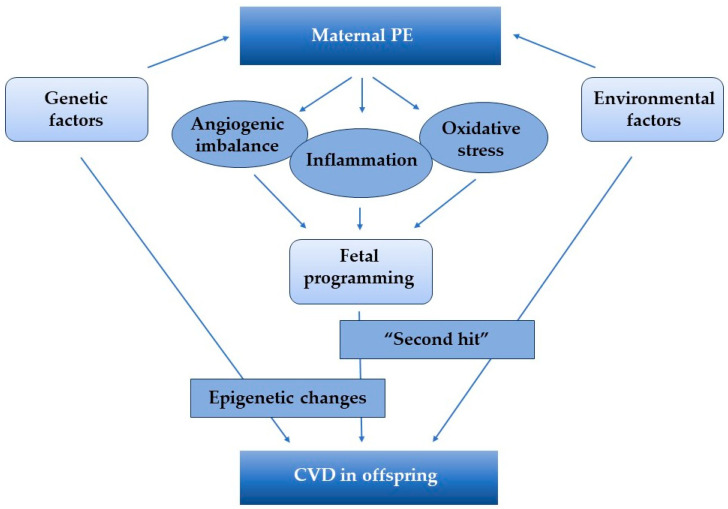
Potential mechanisms associated with in utero exposure to preeclampsia (PE) and cardiovascular (CVD) risk in offspring.

**Table 1 ijms-25-05455-t001:** The results of meta-analysis of Wang et al. [23].

Risk Factors	Patients Exposed to PE	Patients Not Exposed to PE	Results
SBP (mmHg)	3952	42,416	MD = 1.51, 95%CI (1.15–1.88), *p* < 0.00001
DBP (mmHg)	3952	42,416	MD = 1.90, 95%CI (1.69–2.10), *p* < 0.00001
BMI (kg/m^2^)	3920	42,082	MD = 0.42, 95%CI (0.27–0.57), *p* < 0.00001
Total cholesterol (mg/dL)	3257	10,824	MD = 0.11, 95%CI (0.08–0.13), *p* < 0.00001
LDL (mg/dL)	3203	10,441	MD = 0.01, 95%CI (−0.02–0.05), *p* = 0.48
HDL (mg/dL)	3558	36,889	MD = 0.02, 95%CI (0.01–0.03), *p* = 0.0002
Non-HDL	400	26,498	MD = 0.16, 95%CI (0.13–0.19), *p* < 0.00001
Cholesterol (mg/dL)	3549	36,556	MD = −0.02, 95%CI (−0.03–−0.01), *p* < 0.00001
Triglycerides (mg/dL)	3250	10,809	MD = −0.08, 95%CI (−0.09–−0.07), *p* < 0.00001
Glucose (mg/dL)	3173	10,411	MD = −0.21, 95%CI (−0.32–−0.09), *p* = 0.0004

Note: PE, preeclampsia, SBP, systolic blood pressure, DBP, diastolic blood pressure, BMI, body mass index, LDL, low-density lipoprotein cholesterol, HDL, high-density lipoprotein cholesterol, MD, mean difference. If the 95%CI of the combined OR value did not include 1.0, the OR value presented statistical significance at 0.05; if the 95%CI of the combined MD value included 0, the OR value presented no statistical difference at 0.05.

**Table 2 ijms-25-05455-t002:** The results of meta-analysis of Andraweera et al. [24].

Risk Factors	Patients Exposed to PE	Patients Not Exposed to PE	Results
SBP (mmHg)	1559	53,029	MD = 5.17, 95%CI (1.60–8.73), *p* < 0.0001
DBP (mmHg)	1583	52,993	MD = 4.06, 95%CI (0.67–7.44), *p* < 0.0001
BMI (kg/m^2^)	1752	53,293	MD = 0.36, 95%CI (0.04–0.68), *p* < 0.0001
Total cholesterol (mg/dL)	396	3788	MD = 0.47, 95%CI (0.21–1.16), *p* = 0.45
LDL (mg/dL)	258	3465	MD = 0.12, 95%CI (−0.09–0.34), *p* = 0.03
HDL (mg/dL)	503	7684	MD = 0.24, 95%CI (−0.79–0.31), *p* < 0.0001
Non-HDL	306	4058	MD = 0.06, 95%CI (−0.07–0.18), *p* < 0.0001
Cholesterol (mg/dL)	216	1276	MD = 1.33, 95%CI (−1.25–3.90), *p* = 0.83
Triglycerides (mg/dL)	486	4334	MD = 0.01, 95%CI (−0.03–0.05), *p* < 0.0001
Glucose (mg/dL)	215	1276	MD = 0.25, 95%CI (−0.0.03–0.53), *p* = 0.07

Note: PE, preeclampsia, SBP, systolic blood pressure, DBP, diastolic blood pressure, BMI, body mass index, LDL, low-density lipoprotein cholesterol, HDL, high-density lipoprotein cholesterol, MD, mean difference. If the 95%CI of the combined OR value did not include 1.0, the OR value presented statistical significance at 0.05; if the 95%CI of the combined MD value included 0, the OR value presented no statistical difference at 0.05.

**Table 3 ijms-25-05455-t003:** The role of different microRNAs.

miRNA	Expression Level	Target Gene Expression	Outcome
miR-144	Upregulation	Downregulation of VEGFA	Decrease trophoblast viability and proliferation
miR-16	Upregulation	Downregulation of VEGFA	Decrease trophoblast viability, proliferation and invasion
miR-17	Upregulation	Downregulation of VEGFA and HIF1a	Decrease trophoblast viability, proliferation and invasion
miR-20a	Upregulation	Downregulation of VEGFA and HIF1a	Decrease trophoblast viability, proliferation and invasion
miR-195-5p	Upregulation	Upregulation of sFlt-1	Impaired angiogenesis
miR-126	Downregulation	Downregulation of VCAM-1	Decrease in pro-angiogenic factors
miR-155	Upregulation	Downregulation of AT1R	Impaired development for offspring
miR-181a	Upregulation	Upregulation of IL-6 and AT1-AAs	Increased sensitivity for AT1R
miR-1301	Downregulation	Upregulation of IL-6	Increase in AT1-AAs production
miR-155	Upregulation	Downregulation of eNOS	Decreased bioavailability of NO
miR-29b	Upregulation	Downregulation of VEGFA	Decrease in trophoblast invasion
miR-30	Upregulation	Downregulation of IGF-1	Decrease in trophoblast invasion
miR-195	Downregulation	Decrease in TGF-β	Decrease in trophoblast invasion
miR-376c	Downregulation	Decrease in TGF-β	Decrease in trophoblast invasion

## Data Availability

Not applicable.

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
