# Peer review of "The Molecular Basis of the Augmented Cardiovascular Risk in Offspring of Mothers with Hypertensive Disorders of Pregnancy"

_ijms, 2024, doi:10.3390/ijms25105455_

Round 1

Reviewer 1 Report

Comments and Suggestions for Authors

The article presentation titled The molecular basis of the augmented cardiovascular risk in offspring of mother with hypertensive disorders of pregnancy has acceptable features. In addition, the genetic factors mentioned especially under the title of Shared genetic factors seem to be insufficient. It is especially appropriate to expand what is mentioned under this heading.

Reviewer 2 Report

Comments and Suggestions for Authors

Clarify meaning of “Preeclamptic women are usually asymptomatic.” Preeclampsia is only diagnosed once symptoms are present.

What is the importance of prematurity in this context? How does preeclampsia lead to prematurity and low birth weight? While the association is made clear, the reasons for the link i.e. delivery being the only “cure” for preeclampsia should be stated. Sections 2.1 and 2.2 could use some additional information.

What are the significance values for the data presented in Table 1 and Table 2?

Are the studies investigating nephron and cardiomyocyte number done in humans or animals? Would the results be comparable if they are done in humans?

Figure 1 needs to be higher resolution.

Clarify the meaning of “in 31%” at the end of the sentence on lines 161-163.

Are the genetic SNPs found in samples from patients with preeclampsia or the children of patients with preeclampsia? Are there any studies that show a link between SNPs present in mom then also found in child?

Cut one word from “originally often” in the sentence from lines 196-198.

Are intrauterine growth restriction [IUGR] and prematurity exposures like preeclampsia or a result of preeclampsia?

Paragraph discussing preeclampsia genesis from line 226 to line 240 is out of place and should be earlier as part of an introduction to the disease.

There should be a deeper explanation for how epigenetic changes affect placental development and spiral artery remodeling.

Inaccurate to say that sFlt-1 competes with proangiogenic factors, it is a blocker or inhibitor not a competitor.

How does plasma sFlt-1 levels suggest a sex difference in developmental programming? This needs further clarification.

How is the imbalance in vasoconstrictors and vasodilators passed to the fetus? Are the fetal cells producing these factors stimulated?

Are the authors proposing that epigenetic changes during pregnancy are mechanisms for preeclampsia or biomarkers for diagnosis of preeclampsia or both? Their hypothesis on this should be clearly stated.

Are the epigenetic changes and disease pathologies mechanistically linked or just associated? Clarity on the difference between an association/correlation and mechanism throughout the sections on epigenetics is important.

Table 3 titles appear incomplete. Title 1… Title 2… Title 3…

Comments on the Quality of English Language

English quality is mostly good. English language edit will catch a few errors. 

Reviewer 3 Report

Comments and Suggestions for Authors

Dear Authors:

The presented review article entitled: "The molecular basis of the augmented cardiovascular risk in offspring of mother with hypertensive disorders in pregnancy" examines the impact of preeclampsia on the offspring and their future risks regarding cardiometabolic and cardiovascular diseases. A number of risk factors are taken in consideration such as time of birth, birth weight, blood pressure, body mass index, lipid profile, glucose and insulin. Authors state that a complex interplay of genes, environment, developmental programming is a plausible explanation for the development of endothelial dysfunction, that leads to atherosclerosis and CVD. Furthermore, they describe a number of underlying molecular mechanisms like angiogenic imbalance, inflammation, alterations in the renin-angiotensin-aldosterone system, endothelium-derived components, oxidative stress and activation of both the hypothalamic-pituitary-adrenal axis and hypothalamic-pituitary-gonadal axis. Finally, they describe the potential role of epigenetic factors, such as DNA methylation and microRNAs as mediators of these effects and suggest that they could open new avenues for future research and therapeutic interventions.

Comments and suggestions:

The manuscript is very interesting and well written, but there are a few things that need to be addressed.

1.     Check the abbreviations: for ex. preeclampsia should be abbreviated in the text and not only in the tables.

2.     The BMI parameter is still a general useful tool to monitor the patient’s metabolism profile, but the waist circumference is now the best parameter to evaluate the CVD risk. In fact, the adipocytes accumulated in that particular area are called “bad fat” and they are able to produce TNF-a which is a very powerful inflammatory cytokine, linking the adipose tissue with inflammation. A very interesting review (PMID: 34827623) cites an article that specifically addresses this matter (PMID: 32020062) and I think that this issue needs to be mentioned and discussed in the paper in view of these children’s adulthood.

3.     Figure 1 is of really poor quality and needs substantial improvement.

4.     Table 3: title 1, 2 and 3 should be named properly.

Reviewer 4 Report

Comments and Suggestions for Authors

This review takes a simple look at how pregnancies complicated by pre-eclampsia can affect the likelihood of CVD in children born from these pregnancies. However, it could be improved before it is accepted. The main problem is the lack of a summary table of the articles found to demonstrate this link.

1 - item 2.3. Cardiometabolic risk factors is very poorly developed, with only review articles and meta-analyses. I would like to see the original articles cited, rather than a review of existing reviews, which could be more recent....

2- Figure 1 is of poor quality and you can't read what's in the image.

3 - Regarding item 3.2. Shared environment, I was expecting to find the effect of endocrine disruptors as possible inducers of this pathology (e.g. https://doi. org/10.1016/j.mam.2021.101054; https://doi.org/10.1016/j.reprotox.2019.04.004).

4 - The link between this pathology and the concentration of some vasoactive components other than the endothelial ones (e.g. doi: 10.1007/s11906-021-01155-4; doi: 10.1093/biolre/ioz204) is missing.

5 - Regarding androgens, I think it could be more developed, see for example current reviews on this topic.

6 - A summary image is missing.~

7 - the item 6. Future Directions is very good :)

Round 2

Reviewer 4 Report

Comments and Suggestions for Authors

The article is significantly improved, and all the suggestions have been implemented. I recommend accepting the article.

Author Response

We would like to thank you for recognizing our work. Your insights have played a pivotal role in enhancing the quality of our work, and we have truly appreciated the chance to incorporate your comments.